# Numerical and Experimental Validation of Active Vibration Control Logic Performance of a Hybrid Noise Control-Based Brick

Ilaria Ronconi [1,*], Roberta Salierno [2], Ling Liu [3], Andrea Giglio [2], Francesco Ripamonti [3] and Ingrid Paoletti [2]

1 Department of Electronics, Information and Bioengineering, Politecnico di Milano, 20133 Milano, Italy
2 Department of Architecture, Built Environment and Construction Building, Politecnico di Milano, 20133 Milano, Italy
3 Department of Mechanical Engineering, Politecnico di Milano, 20156 Milano, Italy
* Correspondence: ilaria.ronconi@mail.polimi.it

**Abstract:** The limitations of active noise control (ANC) in coping with low frequencies and of passive noise control (PNC) in coping with middle-high frequencies are objects of research that present the potentialities of hybrid noise control (HBC). It aims at combining both of the behaviours by broadening the range of absorbed frequencies. Among the several application fields, the AEC (architecture, engineering, and construction) market can take advantage for those applications in which the noise conditions are caused by sound sources that tune in a broad frequencies range. In this frame, the paper describes the numerical and experimental validation of the active behaviour of an under-development project of a hybrid noise control-based acoustic bricks. The latter intends to embed the potentialities of active vibrational noise control (AVC) and passive destructive interference (PDI) in a unique design of an easy-to-mount, 3D-printed, customisable smart acoustic blocks. Active vibration control, the object of this paper, is provided by a 5-mm thick aluminium circular plate with an attached piezoelectric patch. The vibration of the latter, depending on a specific control law, defines the vibration of the plate itself achieving an abatement of the reflection coefficient. Through mathematical modelling and tests in an impedance tube, the results show that the control logic can reach an average abatement of the reflection coefficient of 82% in the frequency range 144–1007 Hz.

**Keywords:** active vibration control (AVC); plate bending vibrations; acoustic smart structures; aluminum plate; acoustic partitions; piezoelectric patches (PZT)

## 1. Introduction

Active and passive noise control can be seen as complementary techniques in terms of effective frequency attenuation range. Active control attenuates low frequencies more efficiently than passive control which works better at high frequencies [1]. Passive control typically involves the use of damping or mass, while active control uses secondary sources to generate a sound field that interferes destructively with the original source [2]. The successful mass adoption of active noise control has so far been limited to confined spaces such as the interiors of cars and aircraft cabins and headphones [3]. However, the proven effectiveness of active noise control (from here on ANC) in mitigating low-frequency noise is given by physically more 'compact' systems than passive control systems [4]. Nevertheless, since, at the state of this paper, ANC is an energy-based system, its use in high energy consuming field (such as built environment) [5] is still limited.

Traditional passive approaches have been the strategy for noise control. Passive materials achieve control through absorption, diffusion or reflection of sound, which are not mutually exclusive. In order to absorb lower frequencies the material is more effective at higher thickness [6]. This, however, gives two limitations in terms of indoor physical space and costs.

For these reasons, systems with hybrid behaviour have been proposed. Examples of hybrid approaches, involving passive performance enhancement with active control, have been demonstrated on noise barriers, where ANC has been applied to extend the effective height of the barrier by minimising diffracted waves over the top of the barrier [7]. One of the advantages of combining both active and passive approaches in the built environment is its improvement on airflow impact [8]. Active control was firstly implemented to reduce the noise of air flow in ducts and was more recently demonstrated on open windows to sound insulate from outdoor noises [9]. In the context of built environment, unobstructed airflow is important for natural ventilation (NV), an essential requirement of building design to provide public health and to meet the United Nations (UN) Sustainable Development Goals [10].

In this context, the present work shows the development of the active part of a block with hybrid noise control behaviour. The latter, called a smart acoustic block, enhances the hybrid behaviour control by exploiting the potentialities of the active vibrational control (AVC) and of passive destructive interference (PDI) (Figure 1).

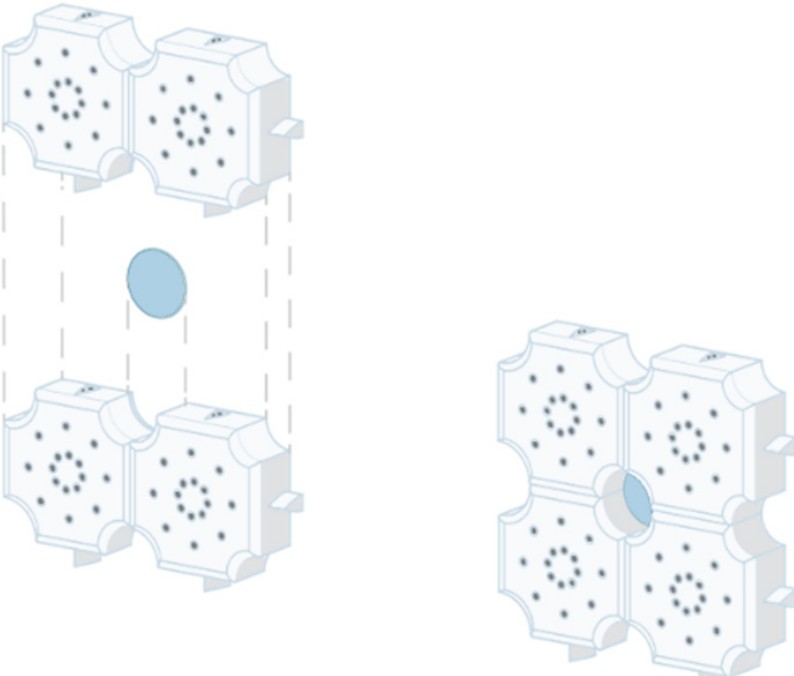

**Figure 1.** Conceptual scheme of the 4-blocks component (with PDI behaviour) that creates the place where to insert the plate (in blue).

In particular, the paper presents the numerical and experimental validation of the active behaviour of the block. In the material and methods section, the system to ascertain the active behaviour and its integration in the block is described. The control law formulation section describes the numerical dissertation on the controlling of the vibration of the plate. The fourth section describes the experimental validation of the numerical. The results and discussions of whole process are presented in the fifth section. The conclusions describe in the last section.

## 2. Materials and Methods

The mechanical system for the active behaviour is compounded by a two mm thick circular aluminium plate with a piezoelectric patch applied on the back.

Piezoelectric transducers are electroacoustic transducer that convert the electrical charges produced pressure variation into energy and vice versa. In our research it is taken in consideration under this last behaviour.

The choice of the circular shape and the radius is given by the 5.3 cm radius of the impedance tube circular section, where the tests are performed (Figure 2).

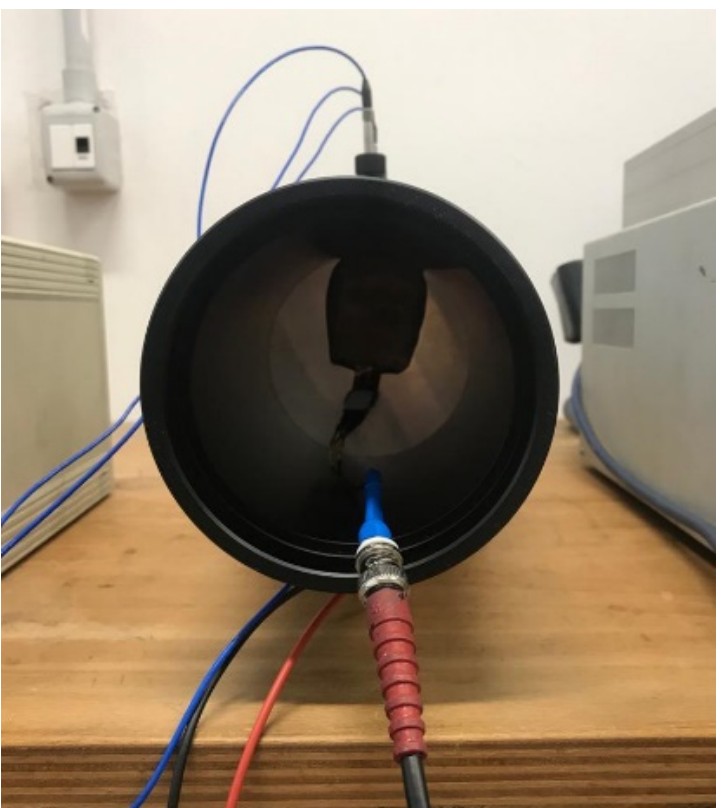

**Figure 2.** Active system compounded by plate and piezoelectric patch in the impedance tube.

Based on that, the model used to study the response of the system plate-patch to the incident sound pressure is a duct, which initially is considered 1 m long by default. The length is reduced after mathematical considerations since it is demonstrated to be less efficient than a lower length.

As far as the radius is concerned, different lengths are not taken in consideration because the chosen size is consistent for the purposes of this research. The condition of the incident plane waves is considered for the formulation of the force to be imposed on the plate [11]. Then, the case of the diffuse sound field is studied. The validation is performed by checking the reduction in the sound pressure wave reflected by the plate [12].

## 3. Control Law Formulation

The active response is given by the vibration of the plate-patch system under the application of the control law. The control law has a key role in controlling the operations of the software code. Generally speaking, the control law has user-adjusted variables and responds to input from the external environment. It acts on its own to perform automated tasks that have been structured into the program [13].

In this case, the control law validation is set up in three steps, starting from a simplification of the case study to a more realistic and complex model.

The first model implements a 2D duct in MATLAB, solved in the frequency domain. MATLAB (an abbreviation of "MATrix LABoratory" [14]) is a proprietary multi-paradigm programming language and numeric computing environment developed by MathWorks. MATLAB allows matrix manipulations, the plotting of functions and data plotting, algorithms implementations, user interfaces creation, and interfacing with programs written in other languages. The rectangular duct sizes are 100 cm long and 10 cm wide.

A uniform harmonic motion ($u_0$) is imposed at boundary $x = 0$ (Figure 3). The control law in terms of vibration velocity is imposed on the other boundary $x = L$. In order to maximise the absorption coefficient of the totally reflective right boundary, a displacement is generated at the controlled boundary:

$$u_L = u_0 e^{-jkL},\tag{1}$$

where $L$ is the length of the tube, $k$ the wave number, $u_0$ the velocity imposed on the other boundary and $j$ the imaginary part.

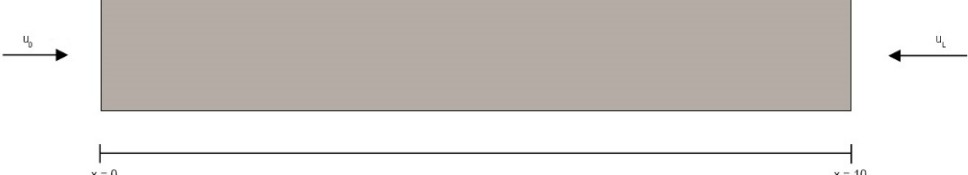

**Figure 3.** 2D duct model.

In this case, a totally absorption of the incident pressure wave is achieved, as shown in Figure 4a, where the sound pressure field is split in incident and reflected contribution:

$$p_i(x) = p_0 e^{-jkx}\tag{2}$$

and

$$p_r = r p_0 e^{-jkx},\tag{3}$$

with arbitrary amplitude $p_0$ and reflection factor $r$.

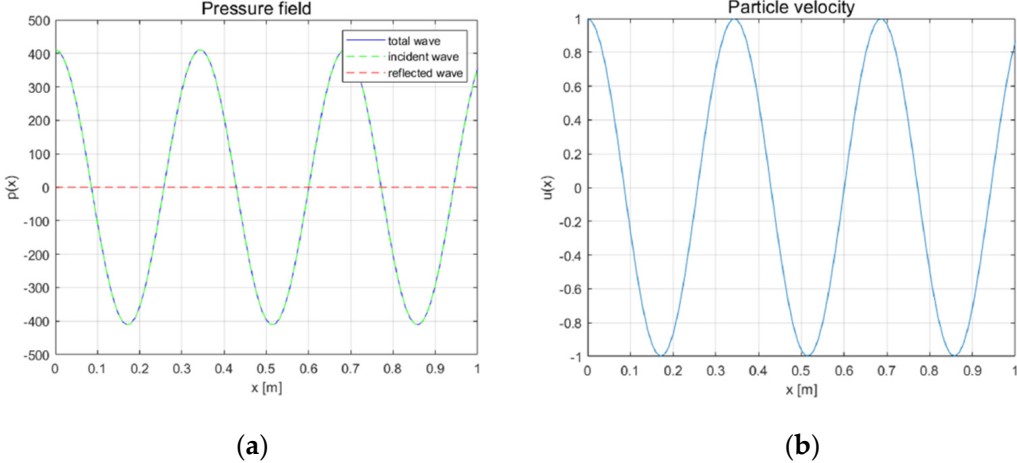

**(a)**                                                                                    **(b)**

**Figure 4.** Closed–closed duct with control law maximising the absorption coefficient as boundary condition in the 2D MATLAB model: (**a**) real part of the pressure field in Pascal and (**b**) real part of the particle velocity field at $f = 1000$ Hz in m/s.

Figure 4b highlights the phase shift of the vibration velocity between the loudspeaker in $x = 0$ and the smart plate $x = L$.

The second model is implemented in COMSOL in order to take in consideration the dynamics of the smart plate.

COMSOL Multiphysics is a cross-platform finite element analysis, solver and multiphysics simulation software. It allows conventional physics-based user interfaces and coupled systems of partial differential equations (PDEs) [15].

Three different variants have been studied:

1. The first one considers only a 2D tube and validates the MATLAB model.
2. The second one considers a 2D simulation including a generic room in which the prototype could be placed instead of the loudspeaker in $x = 0$.
3. The third one considers a 3D tube.

The first 2D variant is necessary to validate the analytical formula implemented for the control law. The second one highlights the applied boundary conditions, validated also in case of diffuse incident field. For the sake of brevity, these two variants are not reported in this paper, but they were necessary to ensure that the MATLAB equation (1) is also valid in the case of a diffuse field and that it is also applicable in COMSOL. The third model considers a 3D tube with a clamped 2-mm thick aluminium shell placed at the $x = L$ position. This makes the model similar to the reality. The COMSOL numerical model evaluate the first resonance of the shell at 1979.6 Hz. This ensures the first mode of vibration. Furthermore, it is verified that the plane wave assumptions are valid up to about 2000 Hz, which corresponds to the first acoustic mode in 3D.

In the third step, the control law is applied as a point load to the plate installed at the end of the tube, simulating the force that is exerted by the piezoelectric patch. A frequency-dependent gain is identified in COMSOL, by considering the frequency response function (FRF) of the shell. The final expression of the point force $F$ in the frequency domain becomes:

$$F = \frac{ju_L}{H(j\omega)} = jH(j\omega)\frac{u0}{\sqrt{2}}e^{-jkL},\tag{4}$$

where $H(j\omega)$ is the FRF of the shell, in terms of the RMS velocity of the entire shell per unitary point force at the centre. Figure 5, as an example, plots the simulation results at 1000 Hz. In particular, the figure plots the incident and the reflected pressure waves are obtained with the transfer function method [12]. The models are tested in the frequency range between 100 Hz and 2000 Hz at intervals of 10 Hz. Figure 5a plots the cancellation of the reflected wave, confirming the ability of the control law in maximising the absorption coefficient.

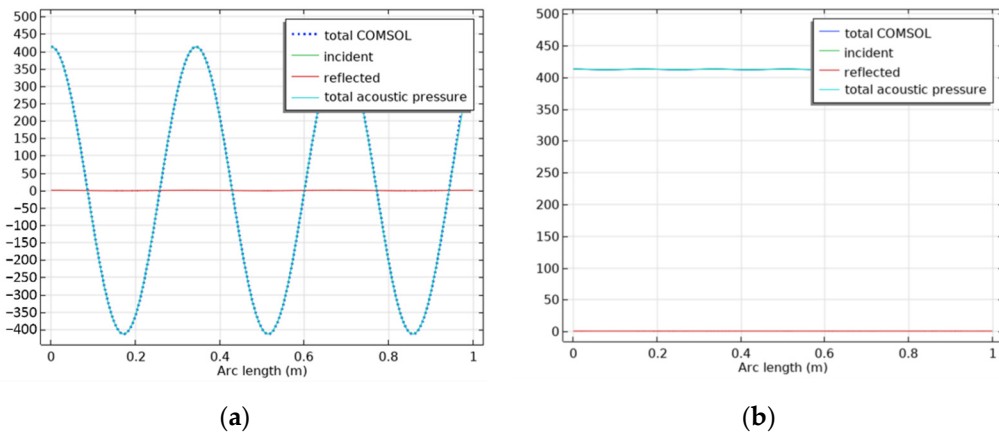

**Figure 5.** Simulation results of the 3D duct model in COMSOL: (**a**) real part of the pressure field in Pascal and (**b**) amplitude of the pressure field at $f$ = 1000 Hz in Pascal.

## 4. Experimental Validation

The experimental validation starts with the verification of the previous model if the point force is replaced with a bending moment.

This step is necessary because the piezoelectric effect of the patch produces a bending moment to the plate [16], as shown in Figure 6a.

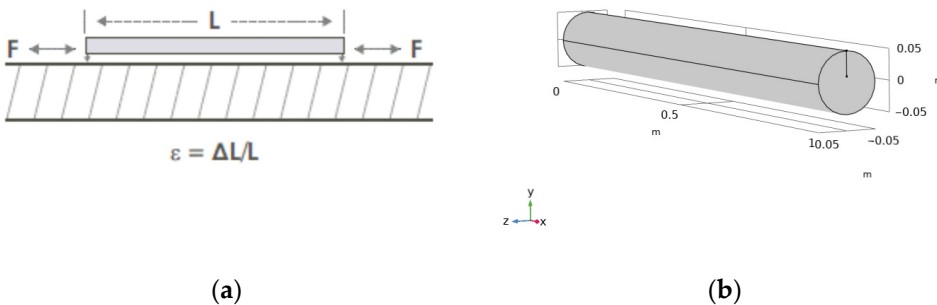

(**a**)                                                    (**b**)

**Figure 6.** Plate-patch system: (**a**) transducer mounted directly to the plate surface. Such a configuration is referred to as a bonded configuration. The bonded configuration is an excellent choice for sensing or creating vibrations in a relatively stiff structure. Transducers in this configuration can be used to monitor vibrations caused by an outside source, section of the plate and (**b**) axonometry of the tube with the applied piezoelectric patch represented by the radius where the moment is applied in the negative z-direction.

Thus, the COMSOL model was updated (Figure 6b). The bending moment per unit length is applied on the line segment in the centre line of the patch:

$$M = M_L e^{j\varphi} \tag{5}$$

The centre line is 4 cm long and it is placed in radial direction of the tube section.

The results at 1000 Hz are presented in Figure 7, demonstrating that the previous control law equation works while also taking in consideration the bending moment. Due to boundary effects, the reflected acoustic pressure is not null but very low.

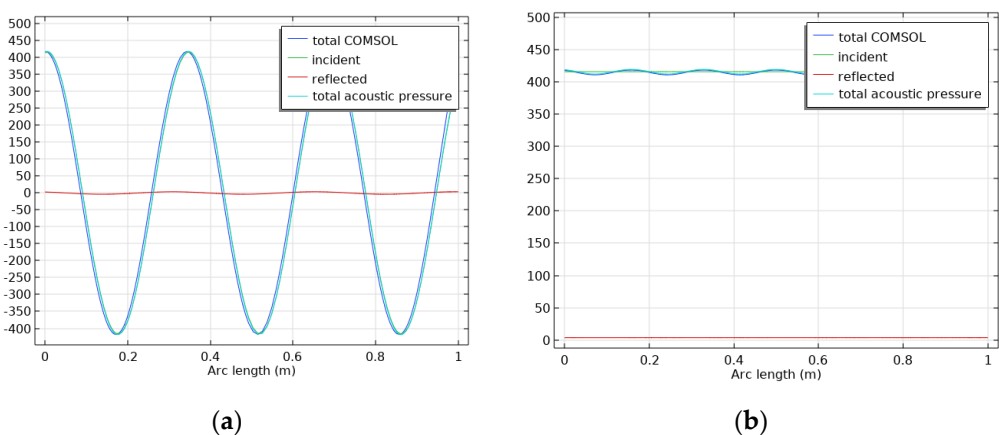

(**a**)                                                    (**b**)

**Figure 7.** Updated model dimension: (**a**) real part of the pressure field in Pascal and (**b**) amplitude of the pressure field at $f$ = 1000 Hz in Pascal.

After this checking and updating the model, several experiments are carried out in the impedance tube.

The first experiment is regarding the validation of the passive case to check that the reflection coefficient of the plate tends to 1. The Impedance Tube (50 Hz–6.4 kHz) Type 4206 of Brüel Kjær was used, which enables the complete measurements such as acoustic impedance and admittance, as well as the coefficients for sound absorption, reflection, and transmission loss.

The tube exploits the two-microphones transfer-function method. A sound source (loudspeaker) is mounted at one end of the impedance tube, in our case correspond to $x = 0$, and a sample of the material is placed at the other end, in our case $x = L$. The loudspeaker generates broadband, stationary random sound waves, which propagate as plane waves in the tube, hit the sample and reflect. The propagation, contact and reflection result in a standing-wave interference pattern due to the superposition of forward- and backward-travelling waves inside the tube. By measuring the sound pressure at two fixed locations and calculating the complex transfer function using a two-channel digital frequency analyser, it is possible to determine the sound absorption and complex reflection coefficients and the normal acoustic impedance of the material [17].

The length of the tube available in the chosen configuration is reduced to 29 cm due to the limitations of the available instrumentation and the radius of the plate on which the sound wave affects is 5 cm.

Figure 8 clarifies the reciprocal connection between instruments in the entire chain set up in the lab to perform the tests.

Microphones (ICP electret model number 130D21) are used for the impedance tube test. A signal conditioner a PCB 8-channel signal conditioner, series 442B, is adopted for conditioning the sensors and an Audio Power Amplifier Type 2716-C was adopted as a power amplifier. During the tests, the sampling frequency is set to $f_s$ = 20,000 Hz and 20 tests are performed, lasting 2 s each. Then, the signals are processed in MATLAB, and the transfer function method is applied to obtain the total, incident and reflected pressure waves in the function of the position inside the tube.

Figure 9 plots the reflective behaviour of the plate and as expected, the reflected wave is symmetrical to the incident one. The reflection coefficient is evaluated for different third octave bands from 100 Hz to 2 kHz, and the results are listed in Table 1.

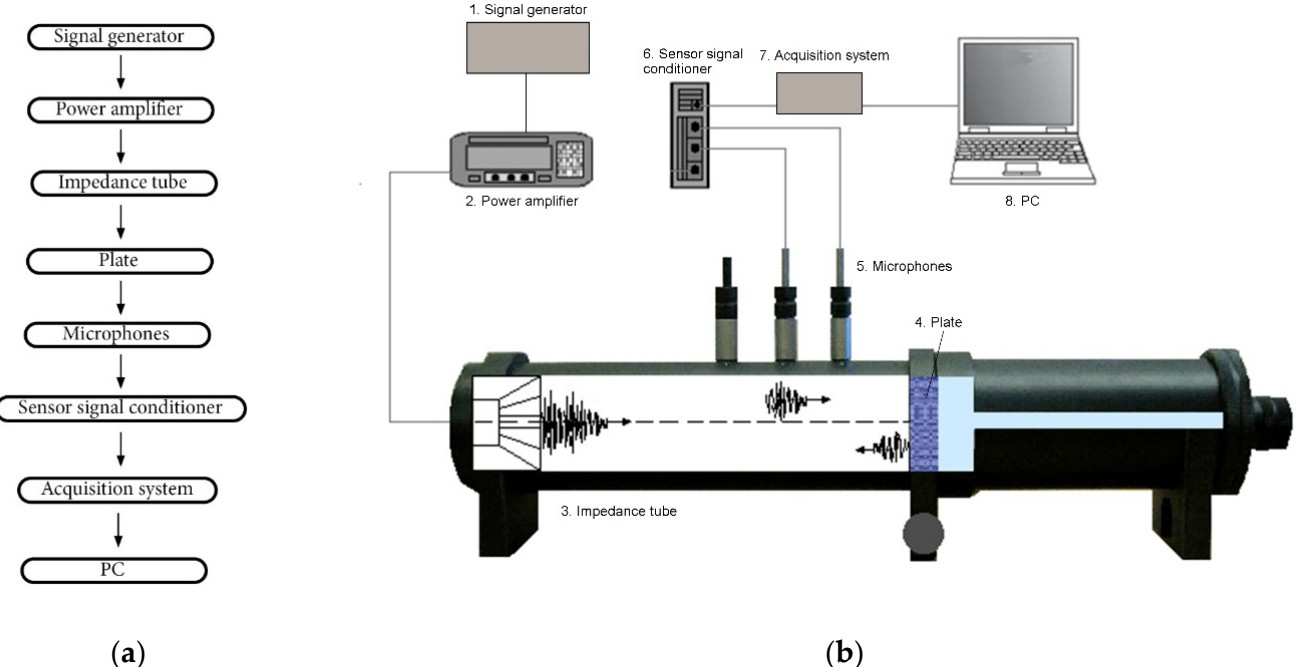

(**a**) (**b**)

**Figure 8.** Setup for tests in the passive case: (**a**) scheme and (**b**) instrumentation.

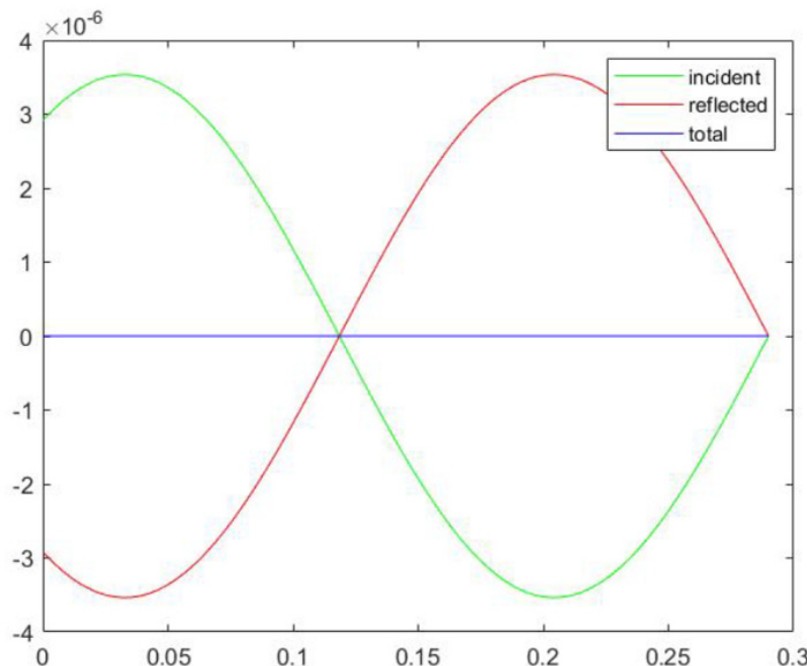

**Figure 9.** Sound pressure reconstructed inside the impedance tube for the passive case at $f$ = 1000 Hz: real part, with acoustic pressure in Pascal in Y axis and tube length in meters in X axis.

For the active case, as previously mentioned, a piezoelectric patch is glued to the plate. The position of the patch is the same of the COMSOL model. The piezoelectric patch is a Midé QP20W Quick-Pack model.

A block diagram for processing the acquired data and realising the active control logic is developed in Simulink.

Simulink is a MATLAB-based graphical programming environment for modelling, simulating and analysing multidomain dynamical systems. Simulink is widely used in automatic control and digital signal processing for multidomain simulation and model-based design [18].

**Table 1.** Reflection coefficient of the aluminium plate in the passive case.

| Frequency [Hz] | Reflection Coefficient |
| :---: | :---: |
| 100 | 1 |
| 125 | 1 |
| 160 | 0.85 |
| 200 | 0.94 |
| 250 | 0.95 |
| 315 | 0.97 |
| 400 | 0.87 |
| 500 | 0.91 |
| 630 | 0.91 |
| 800 | 0.93 |
| 1000 | 0.93 |
| 1250 | 0.93 |
| 1600 | 0.95 |
| 2000 | 0.94 |

The Simulink model is executed by a dSpace Control Board, model DS1006, allowing the control and observation of the system in real time. Thanks to the easy interfacing with MATLAB and Simulink, the implementation of the control logic is feasible, indeed, Simulink provides a graphical editor, customisable block libraries, and solvers for modelling and simulating dynamic systems. It is integrated with MATLAB, enabling you to incorporate MATLAB algorithms into models and export simulation results to MATLAB for further analysis. It is potentiality permitted to simultaneously control and observe the system in real time.

The sampling frequency is reduced to $f_{s,acq}$ = 9 kHz, in order to comply with the performance of the dSpace. Figure 10 plots the schematic setup for the experiment in the active case.

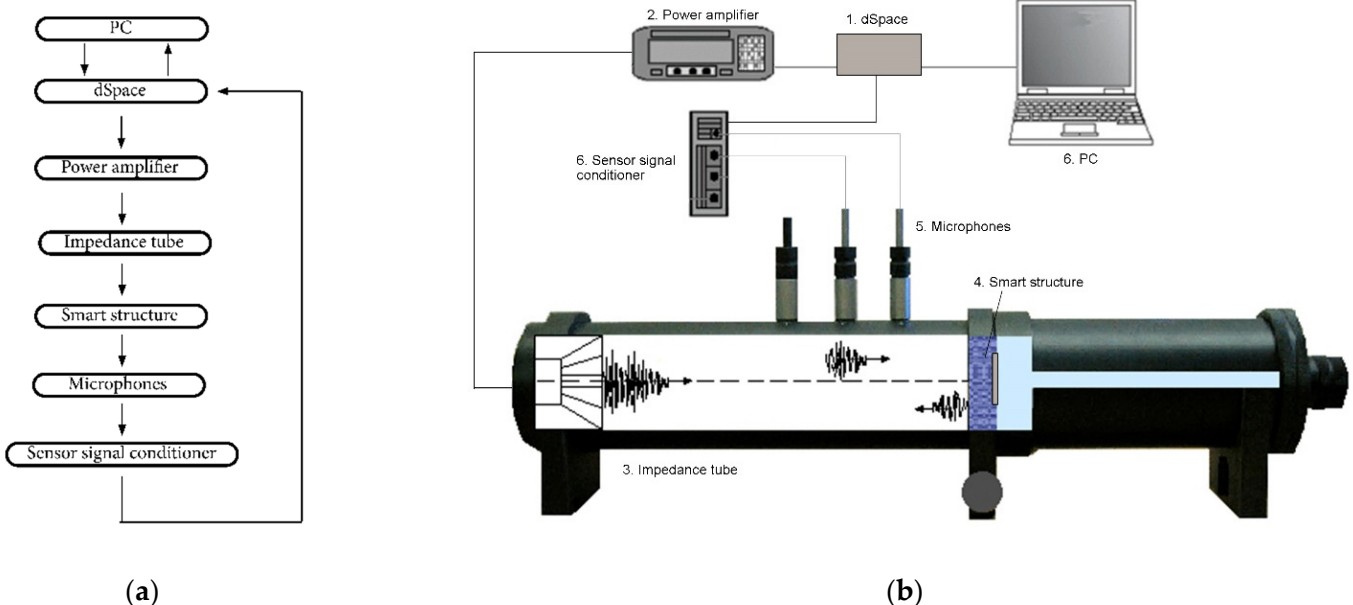

(**a**)                (**b**)

**Figure 10.** Setup for tests in the active case: (**a**) scheme and (**b**) instrumentation.

The dSpace generates two outputs. The first one, connected to the loudspeaker, is a sine wave with amplitude 0.05 and a frequency that can be modified from the ControlDesk application. The second one, connected to the HX amplifier and to the patch, is the control law signal. For inputs, the dSpace receives the sound pressures at the three microphone positions of the tube. The three pressures are exploited to obtain the reflection and incident pressure wave amplitude through the transfer function method. The reflected pressure wave component in the frequency domain is evaluated by

$$A = \frac{p_3}{e^{jkL_{m3}} + re^{-jkL_{m3}}},\tag{6}$$

where $p_3$ is the pressure harmonic component detected at the third microphone position, and $L_{m3}$ the position of the third microphone. The amplitude of this pressure, $|A|$, is used to obtain the incident pressure in $x = 0$, which is given by:

$$p_i = |A|e^{-jk(-L)}, \tag{7}$$

where $L$ is the tube length (29 cm). Consequently, the velocities $u_0$ and $u_L$ are obtained as:

$$u_0 = \frac{p_i}{\rho_0 c}, \tag{8}$$

$$u_L = u_0 e^{-jkL}. \tag{9}$$

Finally, the signal sent to the patch can be expressed as:

$$F = jC(\omega)u_L. \tag{10}$$

where $C(\omega)$ is a real coefficient that depends on the sensitivity of the patch and the frequency response of the plate. Note that since the frequency response of the plate is a function of the frequency various approaches can be used to determine the coefficient $C$. For the purposes of this work, a trial-and-error approach is adopted, and $C(\omega)$ is adjusted in the ControlDesk application to find the optimal result.

In summary, the Simulink model, with its connections to the experimental devices, is reported in Figure 11.

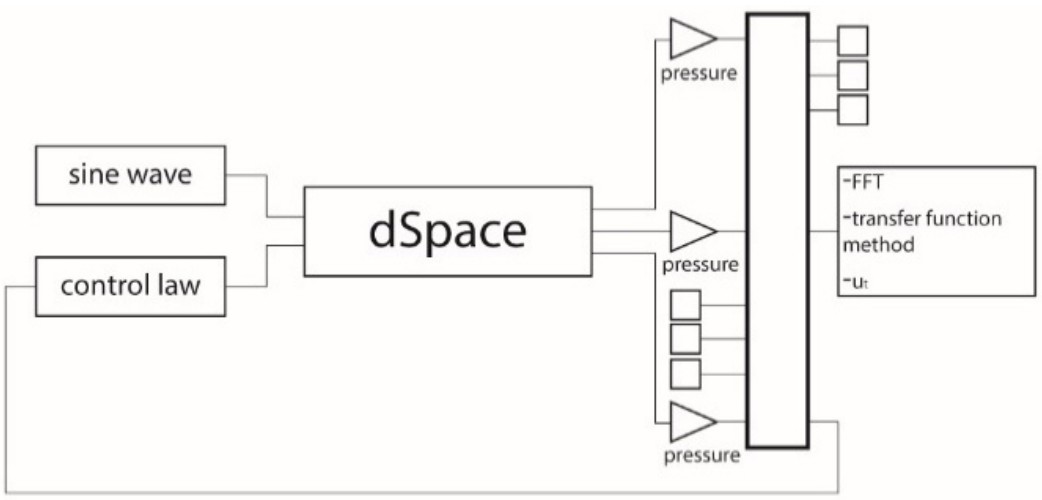

**Figure 11.** Simulink model for the active case.

## 5. Results and Discussion

In this section the experimental results of the control logic that aim at minimising the reflected pressure wave, or in other words, to reduce the reflection coefficient of plate are reported. The expected experimental results show the sound pressure and a particle velocity inside the tube as similar to the ones plotted in Figure 4.

The reflection coefficients obtained after the optimal tuning of the $C(\omega)$ coefficient are shown in the Table 2.

**Table 2.** Experimental reflection coefficients in the active case.

| Frequency [Hz] | Reflection Coefficient | Reduction |
|---|---|---|
| 144 | 0.15 | 85.70% |
| 288 | 0.20 | 79.38% |
| 431 | 0.20 | 79.16% |
| 575 | 0.10 | 89.58% |
| 719 | 0.30 | 73.21% |
| 863 | 0.07 | 92.55% |
| 1007 | 0.15 | 83.51% |

In the experimental validation phase, the results are limited to an upper frequency of 1000 Hz due to the sampling frequency imposed by the dSpace in the real-time system.

They can be compared with the reflection coefficients of the passive case, as shown in Figure 12.

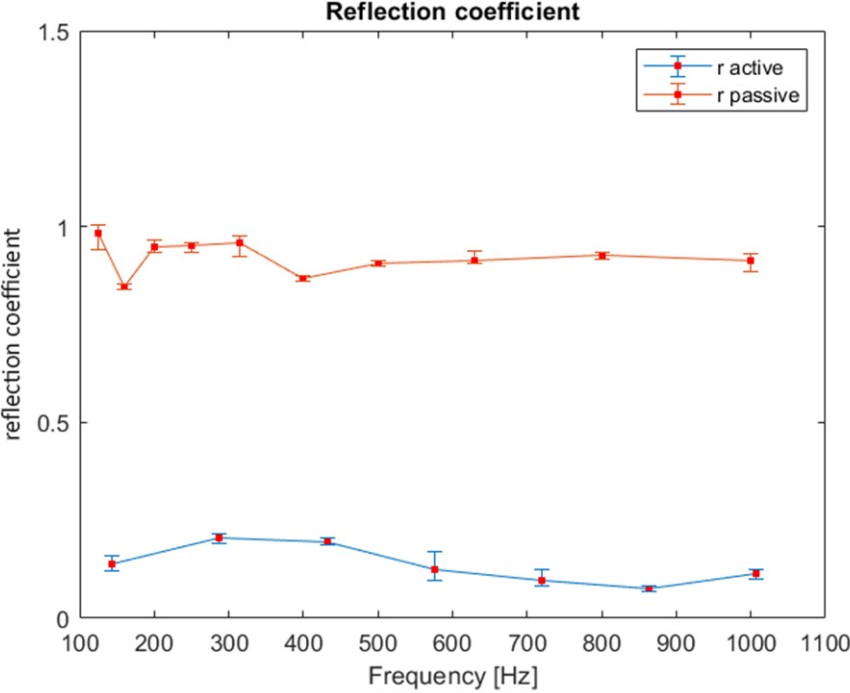

**Figure 12.** Experimental results of the reflection coefficients with their confidence intervals.

In both cases, the tests are repeated ten times, and the mean values and the confidence intervals are reported. An average reduction of 85% and a maximum reduction of 93% at 863 Hz can be appreciated. The sound pressure and the particle velocity fields for the 860 Hz test case are computed from the 3D numerical model and reported in Figure 12. The comparison between the passive and the active case shows in the pressure plot a totally reflected wave for the passive case (Figure 13a) and a cancelled reflected wave for the active case (Figure 13b). For the particle velocity plot the results shows that the boundary conditions for the passive case (Figure 13c) are satisfied and for the active case an increase in the particle velocity on plate boundary (Figure 13d).

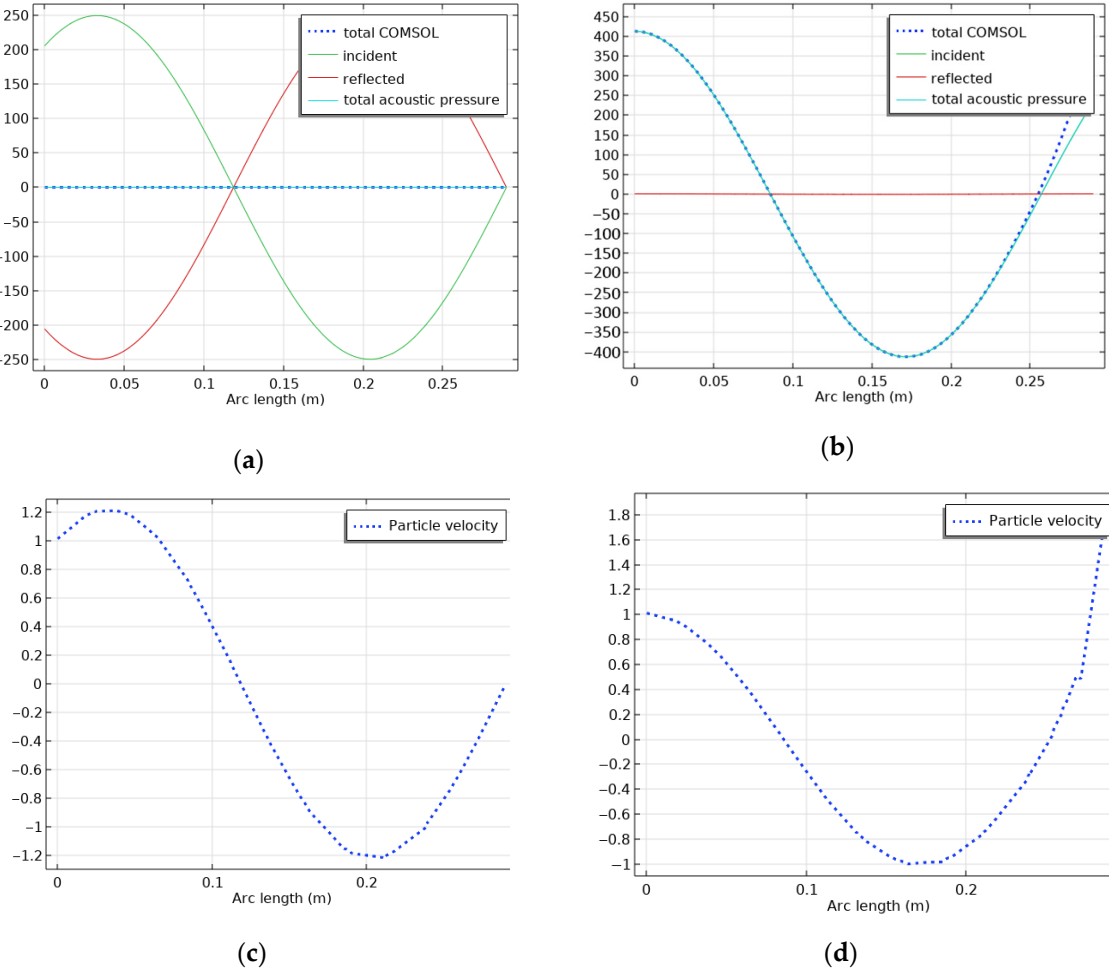

(a)

(b)

(c)

(d)

**Figure 13.** Simulation results of the 3D duct model in COMSOL: (**a**) real part of the pressure field in the passive case in Pascal and (**b**) real part of the pressure field in the active case in Pascal, (**c**) particle velocity of the pressure field at $f$ = 860 Hz in the passive case in m/s and (**d**) particle velocity of the pressure field at $f$ = 860 Hz in the active case in m/s. These results validate the control logic that maximises the absorption coefficients in the low-mid frequencies where the passive solutions are less effective.

## 6. Conclusions

This work presents a numerical and experimental validation of the active part of a brick with hybrid acoustic behaviour to modify its sound absorption properties. The smart block aims to offer a component capable of expanding the frequency range with high sound absorption coefficients. This allows it to be applied in interiors and to more efficiently guarantee high standards of acoustic comfort.

Firstly, the acoustic field in a duct has been analysed and described with an analytical model in MATLAB, where the control logic was formulated. A preliminary evaluation of the results was performed numerically by means of COMSOL, implementing three versions of the system (2D duct, 2D duct in the chamber, 3D duct) for specific analysis of the problem, only the third one was reported in this paper.

In order to validate the control logic, an experimental campaign on an impedance tube was carried out.

Considering the physical limitations of instrumentation and of the smart system, the effectiveness of the control strategy in improving the absorption characteristics of the plate was experimentally proved. The experimental results show that the control logic can reach an average abatement of the reflection coefficient of 82% in the frequency range 144–1007 Hz.

In this paper, the objective was to minimise reflection, but it is possible to apply the same logic to control the absorption coefficient and the reflection coefficient with the objective of both minimising and increasing them. Future works have to be guided towards the integration of the passive behaviour of the smart acoustic block in terms of theoretical validation. After that, the verification of the whole system has to be proven in a real case.

**Author Contributions:** Conceptualisation, A.G.; methodology, F.R.; software, L.L.; validation, I.R. and F.R.; investigation, I.R.; data curation, I.R.; writing—original draft preparation, I.R.; writing—review and editing, A.G., L.L. and F.R.; visualisation, R.S.; supervision, I.P. All authors have read and agreed to the published version of the manuscript.

**Funding:** This research received no external funding.

**Institutional Review Board Statement:** Not applicable.

**Informed Consent Statement:** Not applicable.

**Data Availability Statement:** Data are contained within the article.

**Conflicts of Interest:** The authors declare no conflict of interest.

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
