# Peer review of "Numerical and Experimental Validation of Active Vibration Control Logic Performance of a Hybrid Noise Control-Based Brick"

_acoustics, doi:10.3390/acoustics4030043_

Round 1

Reviewer 1 Report

1. Although the background of the present study is a smart acoustic wall designed for architectural purposed and able to enhance the acoustic experience of interior environments, the main content of this paper is a control logic able to modify the vibration of a plate in order to modify its sound absorption properties, so the current title "smart acoustic wall" is inappropriate. 

2. The Introduction should be rewritten because many recent relevant references aren't included on the active control technique, and the currnt references are old.

3. The control logic to adjust the absorption performance should be describled in detail in a separate section.

Author Response

Dear reviewer,

thanks for your review.

In the attachment you can find an updated versione of the article based on your comments as well.

  1. the concept of "smart acoustic wall" is more defined and explained
  2. more recent references are added to the introductio
  3. the control logic is described in a specific paragraph.

Best.

Reviewer 2 Report

Dear authors,

You have carried out a nice work, the structure of manuscript is correct and the overall purpose of your work motivated. Altogether, I consider it is a valuable and necessary work with an important degree of innovation.

Nevertheless, I feel sad to say that (in my opinion) the research question is not sufficiently informed, some issues in your manuscript should be revised and experimental basis/method be reinforced in order to strengthen findings and research results.

To this end, some comments and suggestions follow:

“Introduction” is too short and limited. You (only) include 8 references in bibliography and use first six in the introduction in order to explain why you accomplish that research, why it is important and relevant. But in doing this, you employ a very old bibliography, old references that in my opinion do not adequately describe the “state of the art” on ANC. Your statements are correct, but its justification not entirely or poorly. Research question is, therefore, not really well defined, correctly explained and justified.

Readers would agree on the convenience of cited research, but would not have a wide and recent view of what is the panorama nowadays. To mention just two, I suggest a quick view to this review paper and follow some of its recommendations: https://doi.org/10.1016/j.buildenv.2021.107928 I would also suggest to check out another MDPI SI "Latest Advances in Active Noise Control" in search of a more recent description on ANC technology nowadays: https://www.mdpi.com/journal/applsci/special_issues/latest_advances_active_noise_control

You comment in “Abstract” that “This work is part of a larger project…” but never talk again about it in manuscript. I suggest to include some more information about this in section 2 (“Materials and Methods”) as the explanations given on the instrumentation and procedure are, in a way, insufficient. The main output of research is a “Smart acoustic plate” which is achieved by “active control” using “piezoelectric patches” and a “control logic” That’s a lot of “questions”, research items that need to be sufficiently explained. Authors should not expect that all readers would be experts on piezoelectric transducers and its applications, how they work and behave or even software experts to understand what’s control logic and how it works. In my opinion, these questions should be explained, at least given the references to be adequately consulted or expanded information, before entering the details of the experimental device.

With regard to experimental device (including a Kundt’s tube), and simulations (MATLAB Simulink and COMSOL) authors should give more information before talking about them as if everybody would be familiar.

Further suggestions insist on this:

A diagram (or a photo) of the “smart structure” is needed in section 2, as Figure 3 in section 3 doesn’t seem sufficient by itself for readers to get an idea of the device.

Reference number [7] dating back to 1996 to justify theory fundamentals seems rather old. “Active control of vibration” fundamentals can be found in much recent books and articles that authors will easily find. Otherwise, please cite the exact assertion that comes from that reference that is not found elsewhere, as reference [7] is a highly and complex theoretical chapter with information that, for sure, exceeds manuscript needs.

Authors are also encouraged to do the same with respect reference number [8], dating back to 1980, as mentioned before.

Section 2 describes the materials, “the model used to study the response of the system”, but readers will easily get lost as authors get to it without giving previous explanations that would help and add clarifications of the procedure.

Authors talk about an “impedance tube” along the manuscript, but give no further reference or explanations on how it works, its basis and limitations, that conditionate their decisions in the experimental design of the research. Authors talk about a “control law” and some other questions before “Experimental validation” arrives in section 3 and go ahead with simulations, but they don’t explain all this either, they are short in the explanations and justifications of what they do.

For example,

what is COMSOL? why COMSOL?

what is Simulink? why Simulink?

what do these simulations give as a result in the research?

Control law??

Where does expression (4) come from??

….

If further explanations are given in section 2, then section 3 would be much easily understood by any interested reader with a medium background on ANC.

Results shown and discussed in section 4 seem rather “small” if we remember the objectives that were initially set, the research question that is also incorporated in the title.

Table 2, “Experimental reflection coefficients in the active case” seems to be the focus point, the main output of research together with Figure 9. But reader would expect “more”, a much deeper analysis of the announced “smart acoustic wall”. Authors are encouraged to incorporate some more details of their innovation, perhaps some other results of the “larger project” in which this research is included.

Considering the above, section 5 “Conclusions” seems too short and somewhat disconnected with text. Without pretending to be offensive, to my best knowledge the manuscript is not self-sufficient. It doesn’t explain itself in a consistent way, any potential reader could get confused if a thorough reading of the text (and perhaps an auto search/learning in some questions) is not carried out. In that case, these questions or similar doubts would arise:

·        This work presents a control logic…” Really? Where? details?

·        The plate is part of a smart acoustic wall….”Acoustic wall? which one?

·        “…the acoustic field in a duct has been analysed and described with an analytical model in MatLab? Which model?

·        …where the control logic was formulated” Is that so?

·        …a preliminary evaluation of the results was performed numerically by means of COMSOL…” COMSOL, numerically, is that so?

·        Considering the physical limitations of the overall architecture…” Which architecture?

·        …control logic can reach an average abatement of the reflection coefficient of 82% in the frequency range 144 - 1007 Hz.” Is that the announced “smart acoustic wall”??

My final suggestion is to revise text in order to include more information on key elements of research (fundamentals and procedure), complete the research with further results from that “larger project” in which this work is included, make it stronger as a research article…..or submit it as a project report or case study or similar entity document, but not a full research article.

Author Response

Dear reviewer,

thanks for your comments.

Based on those, please find in attachment the updated manuscript with the following improvements:

  1. the introduction explained the background of the research adding more recent studies.
  2. more information are provided for the larger project adding a diagram.
  3. more specifications are provided for impedance tube, comsol, etc.
  4. conclusions are expanded taking in considerations also the future development of the whole research and next steps.

Best,

Reviewer 3 Report

This paper designed a active method to reduce noise and absorp the sound. It is appropriate to be published with some modification.

1. The sequential statement repeats itself in the page 3 line98-101

2. Please describe the Y axis and unit used in all Figures.

3. Is there any delay between the incident signal and active signal?

4. The paper shows the result of single frequency, is it possible to work for broadband?

5. Please explain the Figure 3 with more details.

Author Response

Dear reviewer,

thanks for your comments.

In the attachment you can find the updated version of the manuscript based on your comments.

Specifically the new version sees the following updates:

  1. the statment in the page line 98 and 101 are fixes
  2. the y axis and uned in all figures are specified
  3. the paper describes the first stage of a long term research. The validation of the model for broadband will be object of next steps
  4. figure 3 is explained.

Best.

Round 2

Reviewer 2 Report

Dear authors, I think you have done a good job with the correction. Your manuscript reads now much better than before!